# A novel approach to predicting exceptional growth in research

Richard Klavans[1], Kevin W. Boyack[2]*, Dewey A. Murdick[3]

**1** SciTech Strategies, Inc., Wayne, PA, United States of America, **2** SciTech Strategies, Inc., Albuquerque, NM, United States of America, **3** Center for Security and Emerging Technology (CSET), Georgetown University, Washington, DC, United States of America

\* kboyack@mapofscience.com

## Abstract

The prediction of exceptional or surprising growth in research is an issue with deep roots and few practical solutions. In this study, we develop and validate a novel approach to forecasting growth in highly specific research communities. Each research community is represented by a cluster of papers. Multiple indicators were tested, and a composite indicator was created that predicts which research communities will experience exceptional growth over the next three years. The accuracy of this predictor was tested using hundreds of thousands of community-level forecasts and was found to exceed the performance benchmarks established in Intelligence Advanced Research Projects Activity's (IARPA) Foresight Using Scientific Exposition (FUSE) program in six of nine major fields in science. Furthermore, 10 of 11 disciplines within the Computing Technologies field met the benchmarks. Specific detailed forecast examples are given and evaluated, and a critical evaluation of the forecasting approach is also provided.

## Introduction

The prediction of exceptional or surprising growth in research is of keen interest to policy makers in government, military, and commercial organizations [1]. Disruptive scientific and technical innovation generates potential threats and opportunities that can change operating environments. For example, exceptional growth in one research topic can displace another or result in disruptive applications [2, 3]. Anticipating these opportunities and threats is a key element of technical intelligence [4] and strategic planning [5, 6]. In general, more accurate forecasts can better inform resource allocation, investment, and other key decision categories.

Historically, the prediction of exceptional growth in research followed a case study approach. Prior research, such as the National Science Foundation's Technology in Retrospect and Critical Events in Science (TRACES) program in the 1960s, Defense Advanced Research Projects Agency's Topic Detection and Tracking (TDT) program in the 1990s, and IARPA's FUSE program from the early 2010s focused on dozens of areas of research that were relevant to the policy maker. Forecasting methods, when they were used at all, were created and evaluated on a case-by-case basis. A generalizable method for forecasting growth in specific research areas that can be applied at large scale has yet to be accepted.

**Funding:** This study was conducted by employees of SciTech Strategies, Inc. (RK, KWB), a commercial entity, and Georgetown University (DAM), and was funded by the Center for Security and Emerging Technologies, Georgetown University. The funder provided support in the form of salaries for all authors. Author DAM of Georgetown did contribute to the study design and preparation of the manuscript but had no role in data collection or analysis. Results of the study were data-driven and not prescribed in the design.

**Competing interests:** Two authors of this study are employed by SciTech Strategies, Inc., a commercial entity. This does not alter our adherence to PLOS ONE policies on sharing data and materials. Data sufficient to reproduce the main results of the study are provided at figshare (Data DOI: 10.6084/m9.figshare.12241727). Paper-level Scopus data are not freely available but can be licensed from Elsevier.

This study presents a novel approach to the issue of forecasting growth in research. Our approach operates over a model of all possible areas of research—a population of roughly $10^5$ research communities (RC)—and develops indicators that predict whether each RC will (or will not) experience exceptional growth over three-year periods. Three-year growth forecasts (0,1—where "1" denotes exceptional growth) are generated for each RC on a year-by-year basis. The $\sim 10^5$ (0,1) annual forecasts are compared with their (0,1) outcomes. With well over one million separate forecasts, we can evaluate whether specific indicators can meet pre-determined thresholds of forecast accuracy on a year-by-year, field-by-field, or discipline-by-discipline basis. Another novel feature of this study is that forecast accuracy is measured using Critical Success Index (CSI), a metric widely used in weather forecasting [7].

This paper proceeds as follows. First, we provide some background on the identification of emerging topics. We then provide the background on how RC models are created and why we have chosen a specific technique (direct citation analysis) in this study. A general approach for calculating and predicting growth is introduced. Probit analysis is used to identify the lagged indicators that best predict exceptional growth. Forecasts that might be contaminated with future information are identified. Accuracy tests are done across years (2006–2015), using two population models (one created in 2012 and the other created in 2018) and across nine broad fields of research. Specific forecasts in an area of Artificial Intelligence in 2014 and 2018 are provided. The final section focuses on limitations to the method and directions for future research.

## Background

### Identification of emerging topics

The identification or characterization of emergence in science and technology is a subject of continuous and growing interest. A search of abstracts in Scopus for the phrase "emerging technology" returns over 25,000 documents, a tenth of which were published in 2019. The vast majority of these studies are case based, declaring a particular technology to be emerging and then proceeding with characterization. Relatively few studies seek to identify emerging topics a priori using either existing methods or new methods of their own design. A review of the salient literature on methods to identify emerging topics through the early 2010s can be found in Small, Boyack & Klavans [8].

Although most studies of emergence are retrospective [9], forecasting studies do exist. Examples of actual forecasts include work by Daim et al. [10], Bengisu & Nikhili [11], and Zhou et al. [12]. However, even these forecasts are case based, exploring small and well-defined topic areas rather than casting a wide net to forecast emerging events across the entire S&T landscape.

Given the lack of validated methods to identify emerging topics at the time, the FUSE program (https://www.iarpa.gov/index.php/research-programs/fuse) was formally launched in 2011 by IARPA, and ran through 2017. The FUSE Program was a fundamental research program that aimed to see if it was possible to provide validated, early detection of technical emergence that could alert analysts of areas with sufficient explanatory evidence to support further exploration. FUSE was motivated by the need for a forward-looking capability that would support planning by reducing technical surprise with two- to five-year forecasts of related document groups of scientific and patent literature. It sought to capture the "real-world concept of a scientific or technical area or domain of inquiry" (https://www.iarpa.gov/index.php/research-programs/fuse/baa) with indicators that functioned over a wide range of disciplines and technical cultures in English and Chinese.

One author of this paper was the founding FUSE program manager who noted that the primary challenge with the program was finding a robust and defensible way to define and measure performance. Multiple methods were tried over the lifetime of the program, ranging from ranking related document groups by degree of emergence as compared to subject-matter expert opinion to ranking emerging technical terms within defined technical areas as compared to future usage rates.

A wide range of forecast quality metrics and measures was explored, including a specially formulated prominence metric and Mean Absolute Percentage Error (MAPE) calculation, scored by a variety of different formulations of precision, recall, and false positive rate, and ranking performance computations (e.g., Kendall's Tau and Spearman's Rank Correlation Coefficient). Despite multiple pragmatic and research advances (https://scholar.google.com/scholar?hl=en&as_sdt=0%2C47&q=D11PC20152+OR+D11PC20153+OR+D11PC20154+OR+D11PC20155+OR+D11PC2015&btnG=), a number of technical issues were faced in the computation of these metrics. Some of these challenges were associated with changes in small counts swamping growth rate indicators, threshold effects for identifying what is emerging and what is not, and leakage of future information into the data when training the predictive system.

Another significant challenge was finding forecasting methods that were explainable to analysts and decision-makers that would ultimately use the system. After consulting with potential users and others, a heuristic metric was agreed upon. This heuristic is based on predicting extreme weather, a phenomenon that many people are very familiar with. Most people have an intuitive sense that the ability to predict a major storm in three days is extremely difficult. Making a prediction that there is going to be a major storm, and then being right one out of three times, is roughly the state of the art (a 33% true positive and a 67% false positive). Failure to predict a storm that does happen (a false negative) of at a rate of roughly 50% is also state of the art for three-day weather forecasts [13].

The specific indicator we use is Critical Success Index (CSI) and is calculated as TP/(TP+FP+FN), where TP is true positive; FP is false positive; and FN is false negative. The CSI Score for the example provided above is 25%. Given that this is the state of the art in forecasting storms three days out, we have adopted the same threshold for predicting exceptional growth in a research community three years out. The analogy to weather forecasting is very apt in that action may be indicated. Forecasts of bad weather often inspire people to action (e.g., boarding windows, changing travel plans). Similarly, three-year forecasts of exceptional research may present opportunities for action.

For completeness, we note that TP, FP, and FN are used extensively in medical fields, that values of specificity and sensitivity are often calculated and presented, and that this will be the dominant mode of thinking for many readers. However, given that our use case—forecasting of research—has much more in common with weather forecasting than medical testing, we will not make further reference to the medical paradigm.

In this study, we calculate CSI using forecasts of exceptional growth (compared to outcomes) of clusters of documents, or research communities (RC), from our comprehensive, highly granular models of science. Further, this analysis is based on over a million instances rather than on a few examples.

## Comprehensive, detailed models of science

Since we propose to detect exceptional growth in research by looking at the publication growth for a specific RC, the issue of literature clustering (choosing how to partition the literature so that each partition corresponds to an RC and exceptional growth can be detected) becomes

central. The clustering approach used in this study is to identify Kuhnian RCs using the "link-ages among citations" that was recommended by Kuhn [14] but that was not scaled up to clus-ter millions of documents until 2012 with the introduction of the VOS (Visualization of Similarities) clustering methodology by researchers at the Centre for Science and Technology Studies (CWTS) at Leiden University [15]. CWTS has since introduced two major updates to their clustering methodology with the SLM [16] and Leiden algorithms, the latter of which fixes specific problems in the earlier algorithms [17]. The VOS algorithm is thus no longer available.

Among the different ways to use "linkages among citations," we use direct citation analysis as the basis for clustering the documents (and creating a classification system) for several rea-sons. First, it was recommended by Kuhn for very specific reasons. Kuhn did not view RCs as a group of researchers. Rather, each RC was focused on a problem that could be detected by looking at the communication patterns between researchers. According to Kuhn, ". . . one must have recourse to attendance at special conferences, to the distribution of draft man-uscripts. . . and above all, to formal and informal communication networks and in the linkages among citations. . . Typically, it may yield communities of perhaps one hundred members [14, p. 178]." As such, a researcher could be participating in multiple RCs. However, the clustering of researchers does not lend itself to research forecasting. In its stead, citations were a well-known signal of a communication link and were correspondingly recommended as a useful signal for detecting these RCs. Second, it is a first order measure that represents the decisions made by authors about what to cite rather than a second order (co-occurrence) measure. Third, it has been shown to be very accurate as compared to bibliographic coupling and co-citation [18]. Finally, a direct citation computation is tractable. Co-occurrence measures, such as bibliographic coupling, co-citation, or even textual similarity, generate hundreds of billions of links for complete databases such as Scopus or the Web of Science, which makes them com-putationally intractable. Additional information about the history, accuracy, and state of the art of this type of clustering process can be found in Boyack & Klavans [19].

Within the context of creating forecasts of RC growth, this approach—using direct citation information within a set of tens of millions of documents, and clustering using the VOS or Lei-den algorithm—has the following positive features. First, and perhaps most importantly, the type of document clusters that are created using this approach have been shown to represent the way researchers actually organize around research problems, which is a central tenet in Kuhnian theory [18]. Second, document clusters created using this approach are currently being used productively in research evaluation worldwide as part of Elsevier's SciVal tool [20]. Third, indicators can be easily created with very little influence from future information, thereby providing the possibility for testing which indicators are able to predict exceptional growth and allowing others to easily build on this research. Fourth, since the direct citation approach inherently accounts for history, the resulting RCs can be effectively categorized by their stage of growth (e.g., emerging, growing, transitional, mature). One would expect that stage of growth would be extremely important in predicting which RCs will experience excep-tional future growth. Finally, using a model consisting of around 100,000 RCs effectively allows us to test the efficacy and generalizability of different forecasting indicators over different fields of research and time, leading to robust, generalizable results of known accuracy.

## Data and methods

### General approach

The general approach used in this study takes advantage of two separate comprehensive, gran-ular models of science that were created using Scopus data. Each model is comprised of tens of

millions of papers that are partitioned into about 100,000 RCs. These models were created at different time periods (2012 and 2018) using different clustering algorithms. We proceed by:

- Describing how these two models were constructed,

- Defining key terms used through this study,

- Determining the metric for exceptional growth,

- Creating a composite indicator for predicting exceptional growth,

- Testing the accuracy of the composite indicator by model, model age, field, and discipline.

Details on each step are provided below.

## Global models

Two models of science were used in this study. Model one, named DC5, is described in detail in Klavans & Boyack [20]. Briefly, it was created in fall 2013 with the VOS algorithm [15] and an extended direct citation approach [19] using Scopus data from publication years 1996–2012. Data from subsequent publication years through 2017 were added at intervals as updated Scopus data were obtained. Additional papers from 1996–2012 that had been added to Scopus were also added to the model. Table 1 shows the counts by year and when they were added to the DC5 model. Papers were added as follows:

1. For papers with references, each paper was assigned to the RC to which its references had the greatest number of links, and;

2. For papers without references but with an abstract, each paper was assigned to the RC to which it was most related via the BM25 text-relatedness measure [21, 22].

The full DC5 model contains 38.73 million Scopus indexed documents through 2017 assigned to 91,726 RCs.

Model two, named STS5, was created in 2019 using Scopus data from 1996 through May 2019. Thus, it contains the full 2018 publication year, but only a partial 2019 publication year. This model was created using a set of 1.039 billion citation links and the Leiden algorithm [17] and contains 43.28 million Scopus indexed documents through 2018 assigned to 104,677 RCs.

It is important to note that the assignment of papers to RCs in both models relies, to some degree, on future information. For instance, the 2010 papers in the STS5 model were assigned using 69 million references and 39 million citations (from subsequent papers). However, although the contribution from citations is significant, most papers tend to be cited primarily by papers that end up in the same RC. We have run calculations that suggest that less than one percent of papers would move from one RC to another each year due to accrued citations. Thus, while future information does impact the assignment of documents to RCs, that impact seems not to be too severe. This issue would need to be addressed when building a production-level forecasting system.

Table 1 also shows that Scopus continues to add information from previous years to its indexed contents. We assume that the other major citation databases (Web of Science and Dimensions) update contents in a similar fashion, which provides another source of future information that would not have been available when making a forecast in any given year.

## Definition of terms

Most of the terms used in this study are based on an analysis of the publication record of each research community. From these data, one can observe new RCs forming and small RCs

**Table 1. Numbers of papers by year in each global model of science.** For Model 1, numbers of papers added originally and at each update are also shown.

| Year | Model 1 | | | | | Model 2 |
|---|---|---|---|---|---|---|
| | Original | 2016_01 | 2017_05 | 2018_05 | DC5 | STS5 |
| 1996 | 926,967 | 3,747 | 14,008 | 8,739 | 953,461 | 985,021 |
| 1997 | 951,188 | 2,952 | 15,738 | 8,512 | 978,390 | 1,012,744 |
| 1998 | 964,061 | 3,706 | 15,795 | 11,638 | 995,200 | 1,037,234 |
| 1999 | 979,298 | 5,007 | 15,602 | 15,481 | 1,015,388 | 1,054,479 |
| 2000 | 1,031,993 | 11,370 | 23,138 | 13,031 | 1,079,532 | 1,117,072 |
| 2001 | 1,089,015 | 12,670 | 25,912 | 16,467 | 1,144,064 | 1,179,964 |
| 2002 | 1,137,594 | 16,562 | 32,886 | 19,539 | 1,206,581 | 1,247,285 |
| 2003 | 1,207,002 | 41,316 | 19,802 | 17,143 | 1,285,263 | 1,335,419 |
| 2004 | 1,343,284 | 22,680 | 15,439 | 13,735 | 1,395,138 | 1,443,125 |
| 2005 | 1,495,559 | 45,018 | 17,593 | 8,738 | 1,566,908 | 1,623,300 |
| 2006 | 1,606,285 | 43,362 | 16,976 | 10,116 | 1,676,739 | 1,743,001 |
| 2007 | 1,704,068 | 51,182 | 22,055 | 10,151 | 1,787,456 | 1,862,707 |
| 2008 | 1,802,622 | 60,046 | 24,905 | 12,163 | 1,899,736 | 1,977,881 |
| 2009 | 1,919,363 | 70,072 | 20,630 | 11,829 | 2,021,894 | 2,111,872 |
| 2010 | 2,033,280 | 104,847 | 21,567 | 12,284 | 2,171,978 | 2,241,956 |
| 2011 | 2,159,551 | 118,872 | 23,839 | 12,236 | 2,314,498 | 2,393,555 |
| 2012 | 2,169,761 | 182,997 | 43,046 | 21,156 | 2,416,960 | 2,523,847 |
| 2013 | | 2,427,223 | 47,810 | 31,583 | 2,506,616 | 2,622,512 |
| 2014 | | 2,373,740 | 153,262 | 38,794 | 2,565,796 | 2,685,118 |
| 2015 | | 1,688,949 | 754,767 | 5,850 | 2,449,566 | 2,658,829 |
| 2016 | | | 2,380,075 | 155,018 | 2,535,093 | 2,738,674 |
| 2017 | | | 662,165 | 1,742,014 | 2,404,179 | 2,813,466 |
| 2018 | | | | 620,305 | 620,305 | 2,868,636 |

growing. As the publication outputs of an RC become larger, they eventually peak (see Fig 1) in terms of share of worldwide publications in a given year and then lose publication share. RCs can also have a more volatile publication pattern: some grow, peak, lose publication share, and then regain publication share. Each RC has a temporal pattern of publications that can be used to calculate growth and other variables such as vitality. Fig 1 gives an example of the publication pattern of an RC and is useful for defining specific terms that will be used throughout our analysis.

**Forecast year, target year and peak year.** Fig 1 illustrates three concepts we will use in this study. The forecast year (FY) is the year upon which a forecast is based. The target year (TY) is three years after the forecast year. The peak year (PK) is the year of maximum publication share from the perspective of the forecast year. In Fig 1, FY = 2014, and the forecast is made using data up through 2014 only. The peak year (PK) occurred two years before the forecast year.

**Publication share and growth.** In addition, we track relative publication share over time to measure growth in each RC over different periods of time. Publication share is defined as the number of articles in an RC divided by all publications. Growth is based on publication share rather than raw counts to account for annual fluctuations in the overall models due to database growth, ensuring any indicator that might predict exceptional growth cannot be attributed to such fluctuations. Publication share is also desirable from a modelling perspective: it tells us how well the research community is doing vis-à-vis other research communities. The concept of publication share is analogous to the concept of market share.

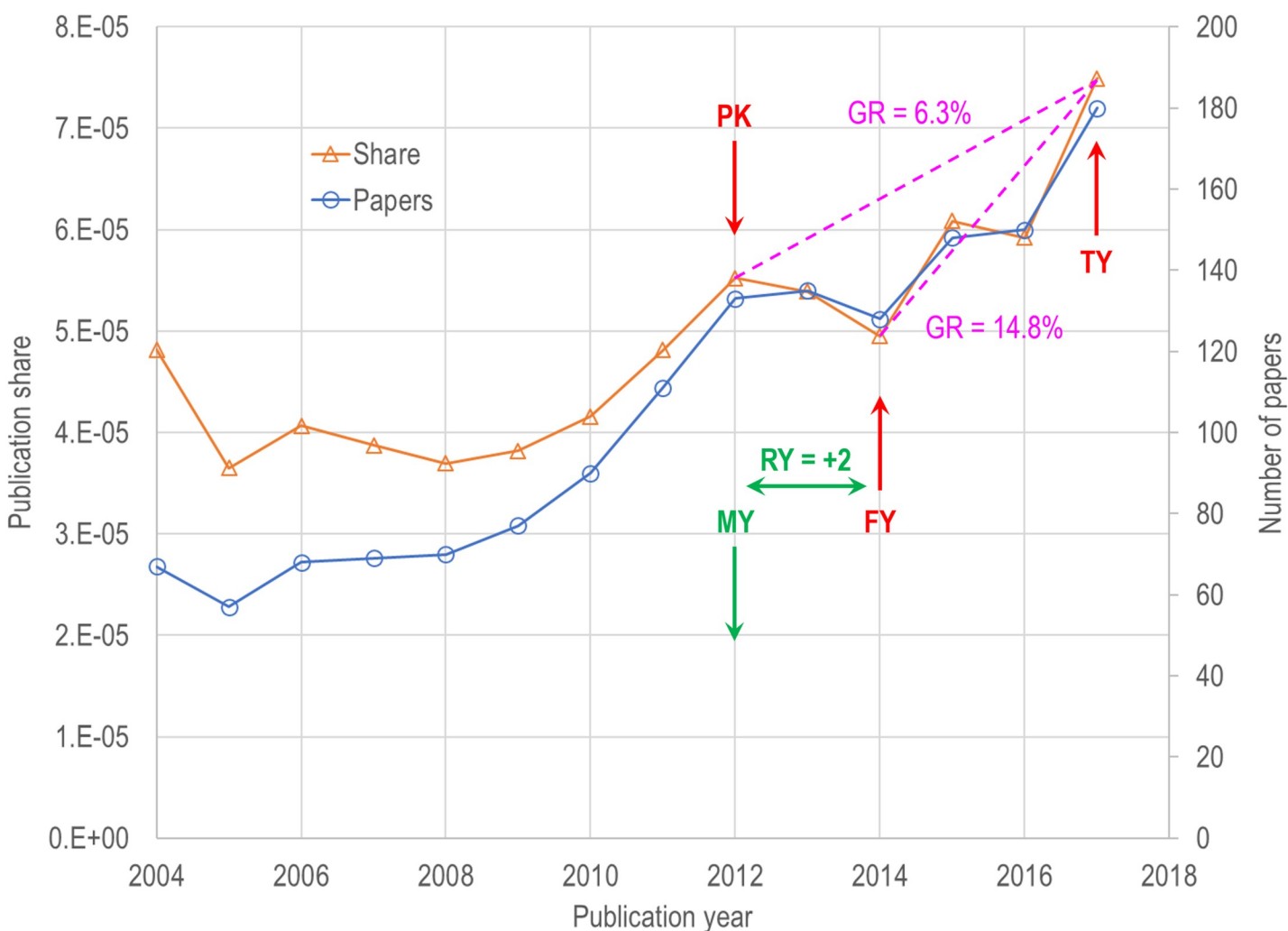

**Fig 1. Temporal profile of a research community in the DC5 model.** The forecast year (FY) is two years after the model was built (MY), and the peak year (PK) occurs before the forecast year. The growth rate (GR) is shown for both the PK to target year (TY) timespan as well as the FY to TY timespan.

The annual compound growth rate for an RC is calculated as

$$\text{GR}_{\text{FY}} = (S_{\text{TY}}/S_{\text{PK}})^{(1.0/(\text{TY}-\text{PK}))} \tag{1}$$

where *S* is publication share, *FY* is the forecast year, and *PK* is the peak year.

Note that growth is measured from the peak year rather than the forecast year. Fig 1 provides an example of a case where they differ. For this RC, the five-year growth rate from the peak is 6.3%. The three-year growth rate of 14.8% overestimates the actual growth due to the publication dip from 2012–2014. This, in essence, delays the signal that a volatile RC might be experiencing exceptional growth and requires that it first has to make up for the dip in publication share. Using our method, the example in Fig 1 does exhibit exceptional growth for FY = 2012 because the growth rate does not exceed the 8% threshold. However, this RC might qualify in FY = 2015 if the three-year growth rate exceeds 8% from 2015–2018.

**Model year and relative year.**   It is extremely important to make the distinction between forecasts that are made before and after a model is created. We have therefore created a variable (relative year, RY) that compares the forecast year (FY) with the year that the model was

built (MY). For example, the relative year for the example in Fig 1 is +2 since FY = 2014 and the DC5 model was built using data through 2012. Note that a forecast can be done for other years on the same RC. For example, for the RC in Fig 1 with FY = 2011, we would have PK = 2011, TY = 2014 and RY = -1.

The reason that relative year is so important is that negative RY have the potential for leakage of future information—i.e., papers in negative RY were placed in clusters using subsequent citations as well as their references. The effect of this future information on the clustering, and thus on forecasts and CSI scores, has not been quantified. Conversely, papers in positive RY were added to clusters without using future information, thus these forecasts have higher integrity than those from negative RY—i.e., they are actionable forecasts.

**Dependent variable–Exceptional growth.** We have defined exceptional growth as a (0,1) variable in order to use precision, recall, and CSI to measure forecast accuracy, with value "1" if $GR_{FY}$ exceeds 1.08 (8%) and value "0" if it does not. While the 8% value may seem arbitrary, it is based on a fifty-year tradition in business portfolio analysis to use a 10% growth threshold (in actual revenue) to distinguish between different market opportunities ('stars' and 'question-marks' are above 10%; 'cash cows' and 'dogs' are below 10%). Since we are using growth in publication share (not actual growth in publication numbers) and the average publication growth rate is roughly 2% per year, using an 8% growth rate in publication share corresponds to a 10% growth rate in actual publication levels. The 8% value is also based on feedback from users. Most users want to evaluate research communities that are opportunities—including those that might be more "on the margin."

## Exceptional growth and relative year

The relationship between relative year and the percentage of RCs that achieve exceptional growth is shown in Fig 2. When all RCs are considered, 5% or more of the RCs in both models have exceptional growth for RY of -3 and lower. However, there is a precipitous drop in the percentage of DC5 RCs that achieve exceptional growth from RY = -3 to RY = -2 and beyond. In contrast, the percentage of RCs that achieve exceptional growth and have at least 20 papers in the FY is relatively constant across models and years (dashed lines in Fig 2) at about 1.5%.

The reason for excluding papers with less than 20 papers is based on user requirements. Users care about potential impact, which can be roughly modeled as mass (the number of papers) times velocity (which corresponds to many of the potential indicators we will be discussing in the next section). A community size of 20 is about the lowest amount that we've observed any user as being concerned about. Below this value there isn't enough mass to make a significant difference in the research environment.

There is a secondary reason for excluding the smaller research communities. We've noticed that there appears to be a large number of very small RCs that only survive for a few years. We are starting to investigate this phenomenon, and it appears that these very small RCs have very sparse networks (very few links between nodes). As we learn more about the relationship between community size, density, and survival, we may modify the size threshold in the future in ways that reflect that learning.

We also point out that small RCs are subject to small number effects. For example, an RC with five papers in the FY only needs to have seven papers in the TY to achieve exceptional growth using our annual 8% growth threshold. The potential bias from small RCs starts to disappear as one gets closer to the year that the model was created. Note that Fig 2 shows that there are relatively few small topics in the DC5 model in RY -2 to +2. This may be due to the inability of the document clustering algorithm to detect small emerging communities with only one or two years of actual history. This might also be due to the fact that when data for

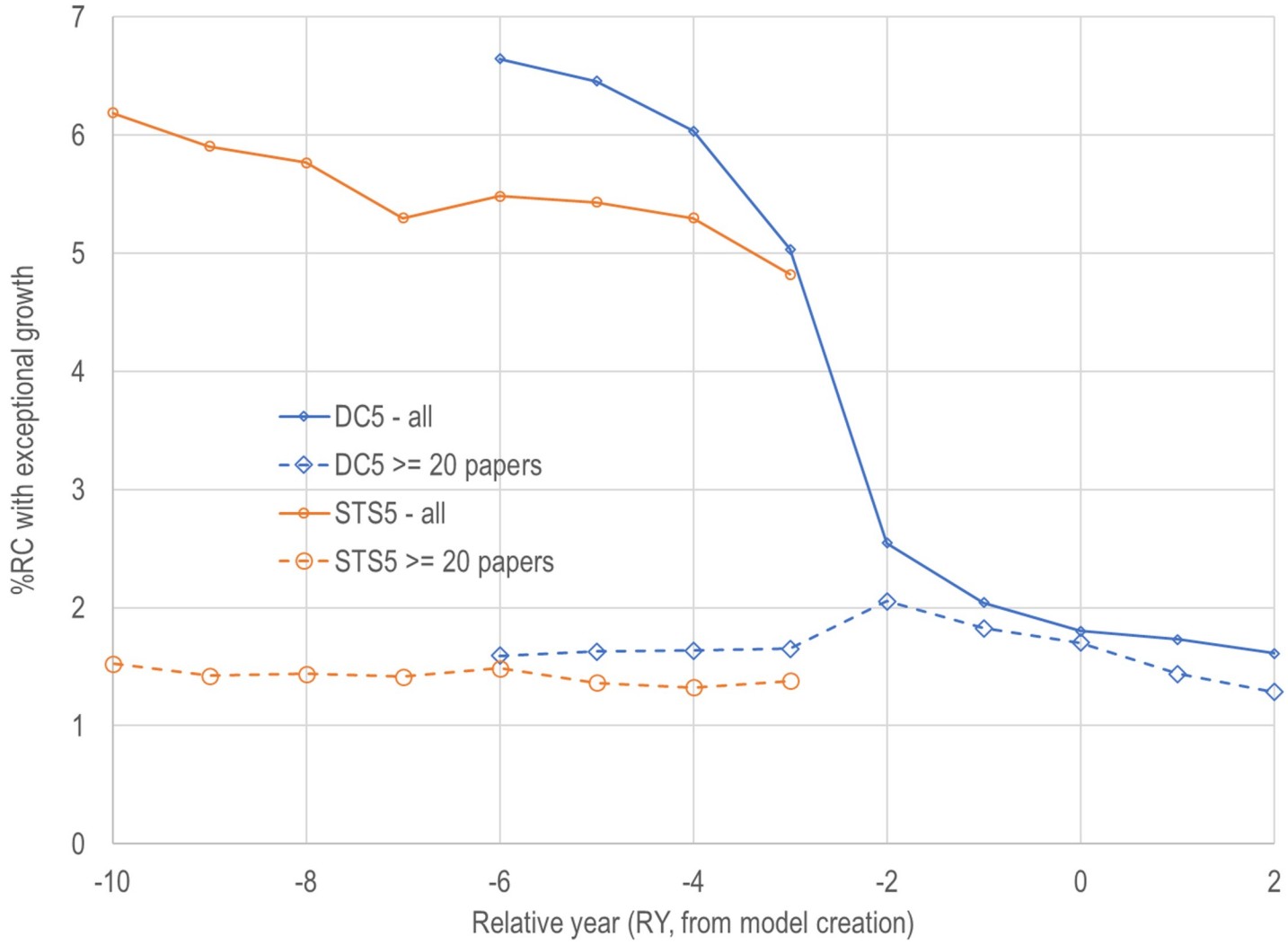

**Fig 2. Likelihood of a research community having exceptional growth before and after a model is created (RY = 0).**

later years were added to the DC5 model, these papers were preferentially assigned to larger RCs, thus limiting the ability for smaller RCs to show growth, and also excluding the possibility for new RCs to be formed in years after the model was created.

Overall, the potential bias and uncertainty introduced by small RCs suggests that future analyses need to be done from two perspectives: 1) using the entire sample and 2) excluding small RCs before a model is built but including them after. The exclusion of small RCs after a model is built is, however, useful from a policy perspective since a community with only seven papers is likely not of sufficient size to have an impact.

## Predictive indicators of exceptional growth

We now turn to the prediction of exceptional growth. Our selection of potential indicators draws from an underlying theoretical assumption that the landscape of research is composed of Kuhnian research communities (RCs). Births and deaths among RCs are not common events when compared to the total number of RCs. Rather, they utilize new discoveries and methods to address an underlying problem that is defined by the community. For example, in

2004 when a scalable method to make graphene was discovered [23], multiple RCs working with graphite (instead of graphene) were already in place that could take quick advantage of that breakthrough. In addition, over the next several years, large numbers of researchers shifted their research to graphene-related RCs, migrating from existing RCs, many of which started to decline as research on graphene emerged and grew. The RCs that supplied the largest numbers of graphene researchers were inherently related to graphene, and included research on carbon nanotubes, single crystals, and electronic properties [19]. Later on, new RCs did form around applications that used graphene (e.g. batteries), but again were populated with existing researchers who had the infrastructure to quickly shift their research focus.

Given this theoretical framework, indicators that reflect the characteristics of a research community were emphasized. Overall, we investigated four variables dealing with the life cycle of the research community, three dealing with assessments of academic importance and three dealing with community size (see Table 2).

**Stage.** Referring to Fig 1, stage is related to the difference between peak year and forecast year and is calculated as (1 / (FY-PY+1). This indicator can be used to estimate the stage of growth of a research community. The longer the time from the forecast year to the peak year, the longer it has been since there has been a significant contribution that resulted in a resurgence of publication. An RC is more likely to be in an early stage of growth if the peak year equals the forecast year and more likely to be mature as the gap increases.

Table 3 shows that this formulation helps to linearize the relationship between stage and the likelihood of exceptional growth. For RCs that are larger (at least 20 papers in the FY) and where the peak year equals the forecast year, 18% and 23% of RCs in the DC5 and STS5 models had exceptional growth, respectively. This percentage drops rapidly in both models as stage decreases. Larger RCs rarely experience exceptional growth if the difference between the peak year and forecast year is greater than three years.

As a further test that the gap between the forecast year and peak year is a valid indicator of stage of growth, we looked at the possibility that the research community would reach its peak in the following year as a function of stage. The results from this analysis are provided in Table 4.

Table 4 focuses on the year right before the two models were built (RY = -1). For the DC5 model, there were 10,663 RCs as of 2011 that had their peak publication year in 2011. Of these

**Table 2. Indicators that were tested for prediction of exceptional growth (std = standardized; log = log transformed).**

| Type | Name | Definition | Transform |
|---|---|---|---|
| Life cycle | | | |
| | stage | Reciprocal length of time to peak year | Std |
| | cvit | Average reciprocal paper age | Std [log] |
| | rvit | Average reciprocal reference age from papers in FY | Std [4th root] |
| | Δrvit | Change in *rvit* over time | See text |
| Academic Importance | | | |
| | ntopj | Number of articles in top 250 journals in FY | Std [log] |
| | ctopj | Number of references to top 250 journals from articles in FY | Std [log] |
| | eigen | Number of articles in top 250 Eigenvalue journals in FY | Std [log] |
| Size | | | |
| | nart | Number of non-review articles in FY | Std [log] |
| | nrev | Number of review articles in FY | Std [log] |
| | nref | Number of references | Std [log] |

**Table 3. Likelihood of exceptional growth (xg) by stage using RCs with at least 20 papers in the FY.**

| FY-PK | Stage | DC5 (MY = 2012, RY = +1) | | | STS5 (MY = 2018, RY = -3) | | |
|---|---|---|---|---|---|---|---|
| | | #RC (2013) | #xg | %xg | #RC (2015) | #xg | %xg |
| 0 | 1.000 | 5,397 | 967 | 17.92 | 4,585 | 1050 | 22.90 |
| 1 | 0.500 | 2,379 | 159 | 6.68 | 2,155 | 170 | 7.89 |
| 2 | 0.333 | 1,814 | 29 | 1.60 | 1,558 | 48 | 3.08 |
| 3 | 0.250 | 1,555 | 10 | 0.64 | 1,354 | 10 | 0.74 |
| 4 | 0.200 | 1,464 | 6 | 0.41 | 1,258 | 3 | 0.24 |
| 5 | 0.166 | 1,543 | 2 | 0.13 | 1,260 | 1 | 0.08 |
| <5 | 0.143 | 13,150 | 4 | 0.03 | 10,730 | 4 | 0.04 |

RCs, 31% continue to increase their publication share in the next year (2012). At the other extreme are the 47,532 RCs that had a peak publication share prior to 2006 (FY-PK < -5). Only 3.9% of these RCs bounce back and achieve a new maximum publication share in 2012. An analysis of the STS5 model shows very similar characteristics. Almost the same number of RCs were clearly in their growth stage in 2017 and had the same likelihood of achieving a new maximum in the next year. Almost the same percentage of RCs were extremely mature and had the same (much smaller) likelihood of achieving a new peak publication level in the next year.

**Current paper vitality (cvit)** is defined as the average reciprocal age of all documents in the RC for a period of time ten years back from the forecasting year. This provides a more nuanced view of when publications have occurred over time. Reciprocal age (1/age+1) is used for much the same reason as above. The "distance" between an article published five years ago versus six years ago is not the same as the "distance" between an article published this year and last year. Use of reciprocal age discounts time so that more emphasis is placed on recent publications and the impact of much older papers is minimized. The natural range of *cvit* is from 1/11 (all papers were published in FY-10) to 1.0 (all papers are published in FY).

We expected (and find) that this variable is highly correlated with *stage*. Whether one (or both) indicators are used will depend on their complementary ability to predict exceptional growth.

**Reference vitality (rvit)** looks at the tendency for researchers to build upon older or more recent discoveries [24]. This is detected by calculating the average age of the references in the papers that are being currently published. In an RC where one may have dozens of papers and hundreds of references, the age of the references tells us whether current activity is building on recent (versus older) literature. If a research community is emerging, there is less prior art and the average reference age will be younger. 1/age is used as a transform, in a similar fashion as

**Table 4. Relationship between stage and likelihood of reaching a peak publication share in the next year.**

| FY-PK | DC5 (MY = 2012, RY = -1) | | | STS5 (MY = 2018, RY = -1) | | |
|---|---|---|---|---|---|---|
| | #RC (2011) | %RC | %pk (2012) | #RC (2017) | %RC | %pk (2018) |
| 0 | 10,663 | 11.9 | 31.4 | 10,491 | 11.4 | 31.1 |
| 1 | 7,884 | 8.8 | 21.2 | 7,628 | 8.3 | 21.1 |
| 2 | 6,503 | 7.3 | 13.8 | 6,569 | 7.1 | 14.6 |
| 3 | 5,647 | 6.3 | 10.7 | 5,669 | 6.1 | 10.7 |
| 4 | 5,579 | 6.2 | 8.5 | 6,528 | 7.1 | 8.5 |
| 5 | 5,744 | 6.4 | 6.2 | 6,586 | 7.1 | 7.0 |
| >5 | 47,532 | 53.1 | 3.9 | 48,905 | 52.9 | 3.9 |

*cvit*, because differences are more pronounced if the references are recent. The variable is normalized using a fourth root instead of a log value, which makes the variable symmetric but with extremely long distribution tails. These extreme values are set at a maximum of +/- 3 standard deviations.

**Change in reference vitality (Δrvit)** is based on the historical change in *rvit*. This indicator is specifically designed to evaluate whether a mature RC has made recent discoveries that shifts the referencing behavior to more recent work. Ten years of *rvit* are used to establish a within-community mean and standard deviation. The 10-year mean *rvit* is then subtracted from the FY *rvit* and divided by the standard deviation to get the difference in terms of numbers of standard deviations (Z score). Since there can be small number effects that give extreme values, these Zscores are bounded by +/-5 standard deviations from the mean.

**Academic importance (ntopj, ctopj, eigen).**    The next three indicators focus on the decisions by editors and reviewers in the top ranked journals to publish articles on a particular topic. These indicators were inspired by the claim that atypical combinations of journals result in higher impact [25]. When we replicated this work, we found that most of the "atypical" citation impact was due to a relatively small number of extremely influential journals [26]. Thus, we decided to test indicators based on papers from these high impact journals to see if they were predictive of growth. *Papers in top journals (ntopj)* counts the number of papers in the top 250 journals as measured by Elsevier's CiteScore. *Citations to top journals (ctopj)* counts the number of references to papers in the top 250 journals—this is closer to the indicator proposed by Uzzi et al. *Papers in top Eigenvalue journals (eigen)* uses Eigenfactor [27] rather than CiteScore to identify the top 250 journals. All three indicators focus on articles (or references) in the forecasting year.

**Size (nart; nrev; nref).**    The final three variables were related to size—*number of articles (nart)* in the forecasting year (excluding reviews), *number of reviews (nrev)* in the forecasting year, and *number of references (nref)* from the papers in the forecasting year. The first two (*nart* and *nref*) focus on community activity (the number of documents in a forecasting year). *Nref* is an indicator of the number of links between documents. A relationship between size and exceptional growth, however, is not expected if the variables associated with life cycle and academic importance are taken into account.

**Transforms.**    Seven of the indicators in Table 4 were log-transformed (i.e., log(value)) because of skewness. This is a common transform when one is dealing with publication activity and citation data. If an indicator can have a value of zero (which only occurs for the size and impact indicators), we use log(value+1). The inverse age transform (1/(year+1) was used for the three variables where it was more important to pick up changes that had recently occurred. The transform used for reference vitality was the fourth root and was specifically designed to create a symmetric distribution since a log transform created a highly asymmetric distribution.

**Standardization.**    After the indicators were transformed, standardization was done by year using the transform (value-mean)/stdev so that the mean and standard deviations would be consistent across years and across models. The use of standardized values across years allows us to combine datasets with different yearly slices of data. This also helps in replication —anyone replicating this work need only standardize their variables and use the recommended coefficients.

## Composite indicator

The composite indicator was based on multi-stage regression analysis, using probit analysis instead of a linear regression model because the dependent variable is binary. We proceeded by identifying the single most important predictor of exceptional growth using Z-statistics, calculating the residual (unexplained variance), and correlating the residual against all non-

selected variables to identify the next most important predictor. This process was repeated until there was no significant improvement in the model (e.g., the newly added variable had a Z statistic less than 4.0). Choosing a Z statistic of four or more as the mechanism for sequentially adding new variables to the model requires further explanation. We are trying to create a model that is as simple as possible (i.e., it contains the minimum number of variables needed) and as stable as possible (the variables are unlikely to be replaced when new variables are created and tested). To accomplish this, we rejected variables that might have a statistically significant effect but don't have a sizable increase in adjusted R-squared. As stated previously, our goal isn't to create a complex model with very high CSI score. Our goal is to present a robust model that is sufficiently accurate to warrant future improvement.

Our analyses (i.e. the nomination of variables using sequential entry into a probit model) were done using eight different data extracts—four using all RCs in two-year periods, and the other four using RCs with at least 20 papers in the FY for the same time periods. The first data extract is the one on which we plan to base additional analysis. We used two forecast years (2013 and 2014) with positive RY (1 and 2) from the DC5 model. This was chosen as the baseline because the assignments of papers to RCs in these two years did not include any future information. Thus, these two years represent actionable forecasts. Probit analysis was done using data from this set of RCs, with four of the 10 variables from Table 4 being found to contribute significantly to the prediction of exceptional growth. Table 5 lists the coefficients for these four variables. All four indicators associated with life cycle were important—they provide different insights into the stage of growth. Only one of the indicators associated with academic importance is used. These four variables were extremely effective in predicting exceptional growth with a pseudo-$R^2$ [28] of 37%. The indicators of size had a negligible ability to marginally improve the pseudo $R^2$.

Table 5 also lists coefficients, sample sizes and pseudo-$R^2$ values for the other seven data extracts. The other three extracts that used all RCs were for time periods before a model was created. The relationship between exceptional growth and these four variables is similar for all four datasets as is the ordering of importance (stage, cvit, Δrvit and ntopj). However, the coefficients are lower for the other three extracts and the corresponding pseudo-$R^2$ values are also lower (a common occurrence when the overall $R^2$ is lower).

Coefficients are also provided for the same four samples using only those RCs with at least 20 papers in the FY. Coefficients for these subsets are higher in all cases, and the pseudo-$R^2$ values are also higher for all but the true forecast (DC5, 2013–14).

**Table 5. Indicator construction using different data samples.**

| Data Sample | | Coefficients from Probit Analysis | | | | #RCs | Pseudo-$R^2$ |
|---|---|---|---|---|---|---|---|
| Model and FY | RY | stage† | cvit† | Δrvit† | ntopj† | | |
| *All RCs included in the analysis* | | | | | | | |
| *DC5 (2013–14)* | *+1, +2* | *0.292* | *0.473* | *0.100* | *0.113* | *161,660* | *0.3735* |
| DC5 (2008–09) | -3, -4 | 0.235 | 0.524 | 0.069 | 0.015 | 178,641 | 0.2694 |
| STS5 (2014–15) | -3, -4 | 0.185 | 0.561 | 0.073 | 0.059 | 172,795 | 0.2706 |
| STS5 (2008–09) | -9, -10 | 0.236 | 0.414 | 0.030 | 0.069 | 178,897 | 0.2070 |
| *Analysis limited to RCs with 20 or more papers in the FY* | | | | | | | |
| DC5 (2013–14) | +1, +2 | 0.312 | 0.540 | 0.167 | 0.124 | 54,347 | 0.3563 |
| DC5 (2008–09) | -3, -4 | 0.374 | 0.481 | 0.134 | 0.040 | 51,849 | 0.3129 |
| STS5 (2014–15) | -3, -4 | 0.393 | 0.583 | 0.087 | 0.067 | 46,137 | 0.3641 |
| STS5 (2008–09) | -9, -10 | 0.410 | 0.624 | 0.176 | 0.068 | 41,081 | 0.3388 |

† transforms for all variables are listed in Table 4.

Coefficients from the true forecast [0.292; 0.473; 0.100 and 0.113] are used to generate an indicator that can subsequently be used to rank all RCs by model, by year and by discipline as:

$$\text{Score} = 0.292*\text{stage}\, \dagger +0.473*\text{cvit}\, \dagger +0.100*\Delta\text{rvit}\, \dagger +0.113*\text{ntopj}\, \dagger \qquad (2)$$

where the transformed indicators as listed in Table 4 are used.

## Results

### Test #1: CSI score by model and relative year

With a composite indicator in place, we now proceed to measure the accuracy of this method in forecasting RCs by model and FY that will achieve extreme growth. This is done for all RCs and for the subset of RCs with at least 20 papers in the FY. The number of forecasts ($N$) to be made is set at 1.5 times the number of RCs that experienced exceptional growth. This is consistent with the initial requirement that precision exceed 33% and recall exceed 50%. Note that this prediction score does not predict growth rate but is intended to rank RCs.

Using the composite indicator equation described above, the $N$ RCs with the highest indicator scores are selected as forecasts. This results in a simple 2x2 contingency table where we can compare (0,1) forecasts made in year FY to their corresponding (0,1) outcomes (whether these RCs experienced exceptional growth or not) in year TY. Contingency tables were created for each model and FY. The CSI threshold for accuracy, established by FUSE, is 25%.

CSI scores from these contingency tables are shown in Fig 3 as a function of model, relative year and whether all RCs or only RCs with at least 20 papers were included. When all RCs are included, neither model reaches the 25% CSI threshold in any year. When small RCs are excluded, the STS5 model is well above the threshold in all years, while the DC5 model is above the threshold in all relative years except -2, -1, and 0. While the STS5 model is above the threshold, this is only for cases where relative year is less than zero (recall that the STS5 forecasts for RY less than zero are using future information).

The trend in CSI score for the DC5 model (using all RCs and using the subset) is perplexing for RY -2, -1, and 0. While the CSI scores are roughly constant for relative years +1, +2, and -3 and below, the scores dip dramatically between years -2 and 0 for reasons that are not clear. Conversely, the STS5 model CSI scores are increasing as the relative year becomes less negative. However, until data are added to this model, we cannot tell what will happen during the next two relative years (-2 and -1) or when information leakage is no longer an issue.

These patterns raise questions that have not been resolved. Is the DC5 dip due to flaws in the way that articles were added to the DC5 model after it was created? Is this due to the flaws that were found (and repaired) in the VOS algorithm [17]? In support of the first possibility, Fig 2 shows a huge drop in the number of small RCs that had exceptional growth from year -3 to -2. In support of the latter possibility, the Leiden algorithm does a better job of assigning papers to RCs than the older (first generation) VOS algorithm. Our sample calculations show that around 86% of documents in the DC5 model are assigned to their dominant RC, while that number is close to 94% for the STS5 model. The Leiden algorithm fixed some problems associated with the earlier clustering algorithms [17], so this may account for some of the differences.

The balance of the analysis will focus on RCs with at least 20 papers since smaller RCs may introduce biases into the analysis, and since they are too small to provide sufficiently reliable information for policy analysis.

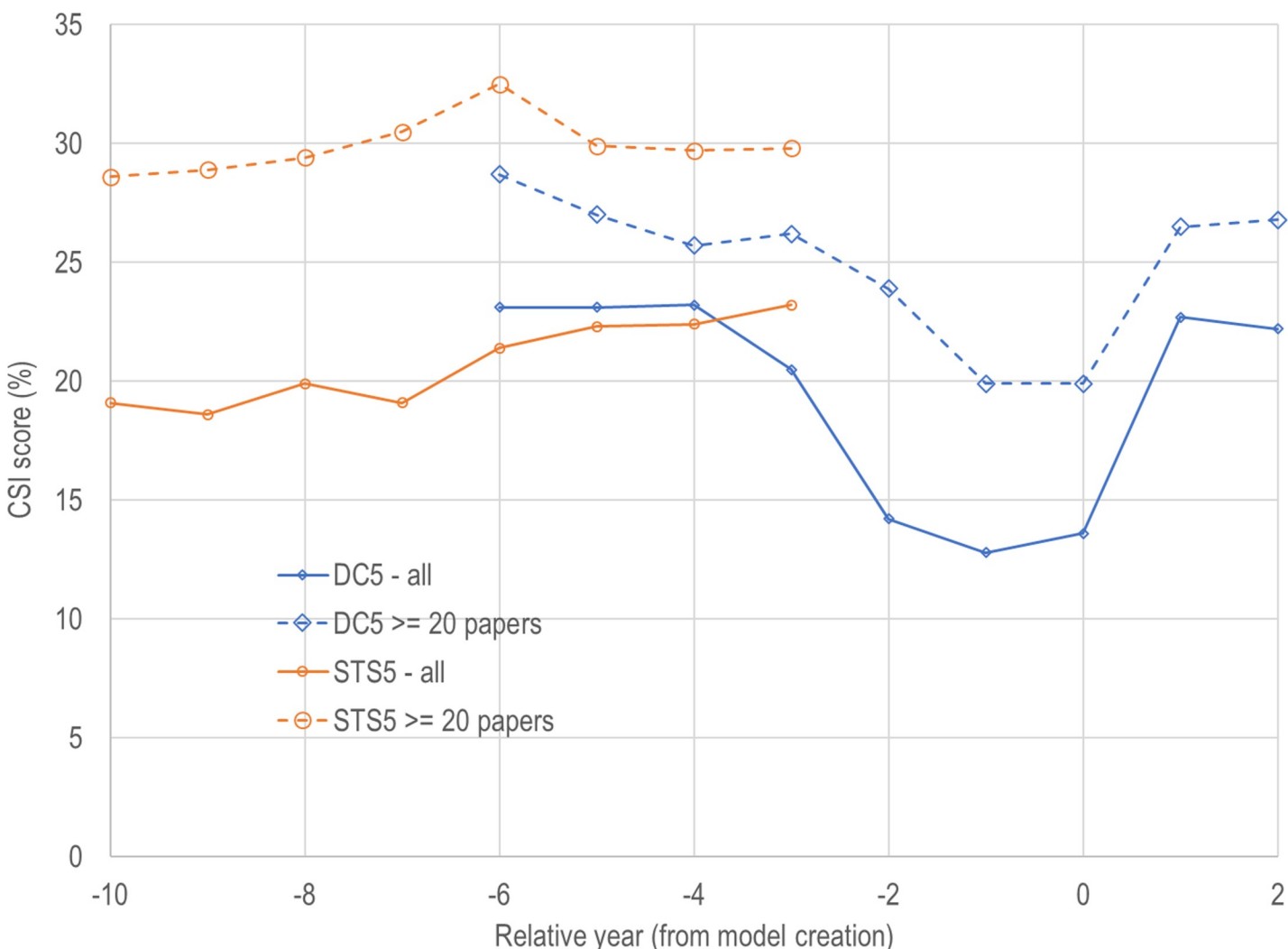

**Fig 3. CSI scores by model and relative year.**

### Test #2: Precision and recall by field (DC5-2014 and STS5-2014)

When restricted to RCs with at least 20 papers, both models have CSI scores that are above the FUSE threshold of 25% in most years. Even though the overall CSI scores are quite high, given field-level differences in citation behavior and characteristics, we expect that performance may differ dramatically by field or discipline. To explore this possibility, we examined performance using groups of RCs aggregated by field and discipline.

The research communities in the DC5 model had been previously aggregated to 114 disciplines (known as DC2 because it has $\sim 10^2$ clusters, while DC5 has $\sim 10^5$ clusters) and then further to nine high level fields. The process by which this was done is described in detail in Klavans & Boyack [29]. While RCs in the DC5 model were directly mapped to disciplines and fields, we assigned RCs in the STS5 models to DC2 disciplines and fields using common papers from 2008–2014.

Table 6 orders these nine fields by non-patent reference (NPR) intensity, which is the average number of times each paper in the field (from 2010–2013) is cited by a U.S. patent (through 2018). Fields at the top of the list (*Biochemistry*, *Computing Technology*, *Applied*

**Table 6. Precision (%prec) and recall (%rec) for nine fields of research.**

| Field | NPR Intensity | DC5 [2014 model year] | | | | STS5 [2014 model year] | | | |
|---|---|---|---|---|---|---|---|---|---|
| | | #RC | #xg | %Prec | %Rec | #RC | #xg | %Prec | %Rec |
| Biochemistry | 0.147 | 2,685 | 98 | *42.1* | *62.2* | 2,321 | 102 | *37.5* | *55.9* |
| Computing Tech | 0.143 | 3,261 | 172 | *42.9* | *64.5* | 3,223 | 253 | *48.2* | *72.7* |
| Applied Physics | 0.125 | 2,451 | 139 | *45.9* | *68.3* | 2,156 | 138 | *44.9* | *67.4* |
| Medicine | 0.099 | 5,466 | 113 | *35.4* | *51.3* | 4,387 | 156 | *39.8* | *59.0* |
| Inf. Disease | 0.077 | 971 | 21 | 31.0 | 42.9 | 803 | 28 | *39.0* | *57.1* |
| Engineering | 0.034 | 2,907 | 163 | *33.9* | *50.3* | 2,915 | 192 | *37.4* | *55.7* |
| Sustainability | 0.032 | 3,618 | 134 | 30.7 | 45.5 | 2,940 | 132 | 27.8 | 41.7 |
| Basic Physics | 0.027 | 877 | 10 | *35.7* | *50.0* | 729 | 17 | 24.0 | 35.3 |
| Civics | 0.015 | 4,473 | 155 | 19.7 | 29.7 | 3,756 | 231 | 30.1 | 45.5 |

*Physics* and *Medicine*) have high precision and recall scores in both models and also have the highest NPR intensities. Research in each of these fields contributes directly to economic development in that it forms the foundation for later patenting and productization. On the other end of the spectrum, two fields near the bottom of the list (*Sustainability* and *Civics*) contribute relatively little to economic development and have much lower forecast precision and recall scores. We find this correlation intriguing, but do not suggest a causal relationship. Rather, these fields of research have inertial properties reflected by the indicators used to forecast growth that are also associated with economic development. In general, the proposed forecasting approach works extremely well in a broad set of fields that have direct economic and health impact.

Two of the three fields with lower precision and recall scores may suggest potential weaknesses in the choice of indicators or even to our choice of theoretical framework. Research in *Civics* (which contains disciplines such as political science, law, economics, and management) and *Sustainability* (which contains disciplines associated with climate change) is easily traced to communities with paradigmatic belief systems. Early indicators of growth (or decline) in these fields might best be picked up using signals from popular media and the internet.

*Basic Physics* had very few exceptional growth events in either model which, while it attests to the steadiness of the field, made this a poor candidate for predicting exceptional growth. This is perhaps not surprising given that this field includes the disciplines of particle physics and astronomy, both of which are dependent on long-term investments in infrastructure such as accelerators and observatories.

## Test #3: Precision and recall by discipline (DC5-2014 and STS5-2014)

The field of *Computing Technology*, along with meeting the FUSE criteria in both models in 2014, has the largest number of RCs with exceptional growth. Table 7 shows that nine of 11 DC2 disciplines in the DC5 model meet the FUSE criteria, while 10 of the 11 DC2s in the STS5 model meet the criteria. There is reasonable correspondence between the two models in that eight of the disciplines meeting these criteria did so in both models. However, there are also differences, particularly in those disciplines that met the criteria in one model and not the other (i.e., *Computing*, *Statistics* and *Mathematics*). These differences may reflect the lack of direct overlap between the way DC2s are reflected in each model, since the DC2s are a direct assignment for the DC5 model and a derivative (matching) assignment for the STS5 model. They may also reflect different dynamics of community behavior or, in the case of *Statistics* and *Mathematics*, disciplines which have relatively few examples of exceptional growth.

**Table 7. Precision (%prec) and recall (%rec) for the eleven DC2 disciplines in the Computing Technology field in both models using the 2014 model year.**

| DC2 discipline | DC5 [2014 model year] | | | | STS5 [2014 model year] | | | |
|---|---|---|---|---|---|---|---|---|
| | #RC | #xg | %Prec | %Rec | #RC | #xg | %Prec | %Rec |
| 9 –Computer Vision/Language | 522 | 43 | *50.8* | *76.7* | 520 | 62 | *52.1* | *79.0* |
| 27 –Networks | 347 | 27 | *51.2* | *77.8* | 340 | 46 | *54.3* | *82.6* |
| 67 –Human Computing | 179 | 19 | *34.5* | *52.6* | 181 | 19 | *44.8* | *68.4* |
| 52 –Telecommunications | 213 | 17 | *38.5* | *58.8* | 199 | 17 | *42.3* | *64.7* |
| 6 –Computing | 560 | 16 | 29.2 | 43.8 | 555 | 47 | *45.1* | *68.1* |
| 34 –Industrial Engineering | 340 | 16 | *41.7* | *62.5* | 340 | 16 | *37.5* | *56.2* |
| 83 –Cryptography | 152 | 12 | *44.4* | *66.7* | 139 | 15 | *54.5* | *80.0* |
| 72 –Statistics | 164 | 6 | *44.4* | *66.7* | 172 | 6 | 22.2 | 33.3 |
| 45 –Operations Research | 240 | 6 | *33.3* | *50.0* | 258 | 10 | *53.3* | *80.0* |
| 102 –Nonlinear Dynamics | 60 | 5 | *42.9* | *60.0* | 57 | 5 | *42.9* | *60.0* |
| 20 –Mathematics | 484 | 5 | 28.6 | 40.0 | 462 | 10 | *46.7* | *70.0* |

## Test #4: Specific forecasts for computing technology

Now that we have established the accuracy of the forecasting methodology for exceptional growth in RCs, we proceed to provide some detailed examples of forecasts for the *Computing Technology* field since it met the FUSE threshold in both models and has the largest number of RCs with exceptional growth. Table 8 lists the top 10 forecasted RCs from the *Computing Technology* field in the DC5 model for a forecast year of 2014. Labels for these RCs are human generated but are based on extracted terms that are highly specific to the RC.

All 10 RCs were at their peak year as of 2014 (the standardized value of Stage is constant at 3.47). All 10 had most of their papers published very recently (current vitality, once standardized, was over 3.3). But the next two standardized variables (change in reference vitality and the number of papers in the top 250 journals) do not provide a consistent signal that these research communities will experience exceptional growth. The four values listed in Table 8 were combined using the coefficients in Eq (2) to generate the score.

Overall, the accuracy of our model is exceptionally good in this field. Eight of the top ten RCs did, in fact, experience exceptional growth. The growth rate of the two RCs that didn't meet the threshold wasn't even close (4.4% and -1.0%). We have not, as yet, analyzed cases where the actual growth rates of RCs that were expected to have exceptional growth were significantly below the 8% threshold.

**Table 8. Top 10 forecasted DC5 RCs from the *Computing Technology* field (FY = 2014, TY = 2017, RY = +2).**

| DC5 | Label | Stage† | Cvit† | ΔRvit† | Ntopj† | Score | Growth |
|---|---|---|---|---|---|---|---|
| 25308 | software defined networks | 3.47 | 5.03 | 0.54 | 3.12 | 3.80 | *45.9%* |
| 48081 | D2D communication | 3.47 | 4.95 | 0.50 | 1.76 | 3.60 | *27.8%* |
| 12007 | mobile security/malware | 3.47 | 4.32 | -0.05 | 2.76 | 3.36 | *11.2%* |
| 14215 | Twitter event detection | 3.47 | 4.45 | -0.24 | 2.32 | 3.36 | *9.3%* |
| 54895 | nature-inspired optimization | 3.47 | 4.50 | 0.77 | 0.97 | 3.33 | *17.9%* |
| 23854 | computation offloading | 3.47 | 3.98 | 1.65 | 2.32 | 3.32 | *34.6%* |
| 14700 | appliance load monitoring | 3.47 | 3.55 | 0.73 | 4.62 | 3.29 | 4.4% |
| 13672 | cellular network energy efficiency | 3.47 | 4.13 | 0.24 | 2.32 | 3.25 | -1.0% |
| 3922 | EV wireless charging | 3.47 | 3.31 | 1.47 | 4.62 | 3.25 | *20.2%* |
| 31270 | internet of things | 3.47 | 4.43 | -0.06 | 0.97 | 3.21 | *54.8%* |

† values listed are after transforms and standardization have been applied

**Table 9. Top 10 forecasted STS5 RCs from the *Computing Technology* field (FY = 2014, TY = 2017, RY = -4, #papers in 2014> = 20).**

| STS5 | Label | #Papers | Score | Growth |
|------|-------|---------|-------|--------|
| 6681 | cloud radio access networks | 126 | 2.65 | *48.7%* |
| 3602 | D2D communication | 377 | 2.64 | *14.0%* |
| 385 | software defined networks | 675 | 2.62 | *24.7%* |
| 7974 | cellular content caching | 75 | 2.61 | *67.5%* |
| 44976 | (general computing) | 80 | 2.60 | -77.0% |
| 4223 | nature-inspired optimization | 247 | 2.60 | *18.5%* |
| 61637 | ontology mapping | 34 | 2.56 | -30.4% |
| 3046 | EM wave metamaterial absorbers | 439 | 2.52 | *10.9%* |
| 24180 | spectrum sharing | 50 | 2.47 | 1.6% |
| 51600 | (general image processing) | 23 | 2.46 | -6.0% |

Table 9 lists the top 10 forecasted RCs from the *Computing Technology* field in the STS5 model for a forecast year of 2014. In this case, the relative year is -4 (the model hadn't been created and all measures are subject to the leakage of future information). We correspondingly included information about the number of papers in 2014 to illustrate the problem of small topics mentioned previously.

One would not actually use the 2014 data slice from the STS5 model for making actionable forecasts for the reasons mentioned previously. But the data in Table 9 do provide insights into the nature of information leakage. The five smallest RCs were only able to predict one out of five cases of exceptional growth. The five largest RCs all had exceptional growth. Stage cannot be used to differentiate these RCs since they were all at their peak publication year. The four RCs with the highest *Ntopj* value had exceptional growth while the remaining six had actual *Ntopj* values of zero and one (with corresponding scores of -0.31 and 1.14 in Table 9).

We also noticed that two of the RCs that were false positives seemed to have more ambiguity in the phrases used to describe the research. Cluster 44976 and 51600 did not have a clear theme. Specific terms extracted from titles and abstracts of the documents (and the papers themselves) in these RCs were only related generally, rather than in a specific way that is common to most RCs. Based on our comparisons of the results in Tables 8 and 9 (and looking at many other RCs from both models), we have a higher level of trust in the actionable forecasts for positive RY than for the forecasts made from negative RY. The data in Table 9 support our initial suspicion that the clustering algorithm may be overestimating the number of small RCs for negative RY. Thematic clarity may be a feature that we should consider as a filter in future studies.

## 2018 forecasts (STS5 model)

Our final step is to provide actionable forecasts based on the STS5 model. The forecast year is 2018. There is no leakage of future information in the creation of these forecasts. Here we focus a little more tightly on a discipline that focuses on Artificial Intelligence applications (DC2 = 9). We will not go over the components of the score—their distribution is similar to what was observed in Tables 8 and 9. Rather, we focus more on who was the research leader in each research community.

The list of top 10 RCs shown in Table 10 forms a very interesting group. Each RC is very well defined with a key phrase. Large, medium and small RCs are all represented. Top institutions in Table 10 are based on activity (number of publications) rather than impact (citations per paper). The most distinctive feature of this list is the large number of industry leaders (four out of 10) and the hegemony of China and the United States, with the top institution in all but

**Table 10. Top 10 forecasted STS5 RCs (FY = 2018) from the *Computing Technology* field.**

| STS5 | Label | #P | Score | Top Institution and Country | |
|---|---|---|---|---|---|
| 5495 | generative adversarial networks | 964 | 3.37 | Alphabet | U.S. |
| 27709 | intelligent fault diagnosis | 142 | 3.27 | Xi'an Jaiotong Univ | China |
| 105 | convolutional neural networks | 4238 | 3.11 | Tsinghua Univ | China |
| 3647 | semantic image segmentation | 1038 | 3.03 | Univ CAS | China |
| 44644 | deep computational models | 58 | 3.00 | Dalian Univ | China |
| 6403 | image captioning | 615 | 2.92 | Microsoft | U.S. |
| 28965 | hate speech detection | 105 | 2.88 | Poly Univ Valencia | Spain |
| 30977 | ReLU networks | 62 | 2.86 | Alphabet | U.S. |
| 37537 | few-shot learning | 53 | 2.85 | Alphabet | U.S. |
| 1005 | word embedding | 1831 | 2.79 | Tsinghua Univ | China |

one RC. Alphabet is the parent company of Google and is the leader in three of the top 10 RCs, while Tsinghua University is the leader in two of the top 10.

Table 10 gives only a sampling of the features that can be used to describe STS5 RCs. Fig 4 shows an example of a characterization of topic #5495, which includes top phrases, phrases that differentiate this topic from others, top categories, journals, institutions, countries, authors, etc. It also shows the temporal history of the topic (document counts per year), a sample of recent papers that are central to the topic in terms of their citation characteristics, and a few top cited historical papers. Finally, a variety of indicators are shown at the bottom right. Characterizations such as these, along with a listing of the papers that comprise the topic, can be used by analysts to understand the history and content of a topic and thus inform policy recommendations and decisions. We look forward to scoring the accuracy of the extreme growth forecast classifications, those in Table 10 along with many others, after data through 2021 are added to the STS5 model.

## Flaws and future directions

The ultimate goal of this project is to create a regularly updated data-driven forecasting system based on automatically generated RCs in all discipline areas. Community characteristics and technology application maturity levels will be continuously measured and forecasted. Changes in forecasts with an auditable method for identifying the source of the change will allow for policy makers and planners to maintain an awareness of how new work is impacting previous assumptions and decisions and will allow them to update judgments as new evidence comes in. There is much work to be done to get to this desired state.

Overall, this study has been extremely helpful toward the accomplishment of this goal. It has introduced a method to forecast which research communities in a highly granular model of science will achieve extreme growth. Although Scopus data were used here, the method can be applied to any comprehensive citation database. This study has also measured forecast accuracy of growth in scientific research using hundreds of thousands of events, a scale which has never before been attempted, much less achieved. The overall results are both reasonable and encouraging. Although the results for the overall models do not meet the FUSE criteria of a CSI score of 25% in every field, they do meet the criteria in fields of particular importance to national security. Gains in accuracy may be achievable with the addition of complementary databases, improvements in the modeling approach, and the development of field specific indicators.

Despite this progress, there are both conceptual and methodological assumptions to this study that need to be viewed from a more critical perspective. From a conceptual perspective,

DC5  5495

CPP1719: 3.856

FIELD: COMP SCI

| TOP PHRASES (2015-2019) | score |
|---|---|
| 1 generative adversarial networks | 0.2219 |
| 2 Experimental results | 0.1219 |
| 3 generative adversarial network | 0.0897 |
| 4 deep learning | 0.0726 |
| 5 machine learning | 0.0628 |
| 6 generative models | 0.0623 |
| 7 Artificial Intelligence | 0.0564 |
| 8 generative model | 0.0550 |
| 9 neural networks | 0.0481 |
| 10 neural network | 0.0466 |

| IDIOSYNCRATIC PHRASES (2015-2019) | score |
|---|---|
| 1 generative adversarial networks | 10.95 |
| 2 generative adversarial network | 3.45 |
| 3 generated images | 2.46 |
| 4 mode collapse | 2.23 |
| 5 image-to-image translation | 2.23 |
| 6 conditional generative adversarial networks | 2.22 |
| 7 style transfer | 2.19 |
| 8 generative adversarial nets | 1.87 |
| 9 adversarial networks | 1.80 |
| 10 image generation | 1.72 |

| TOP CATEGORIES (2015-2019) | #pap |
|---|---|
| 1 Software | 490 |
| 2 Computer Vision and Pattern Recognition | 444 |
| 3 Signal Processing | 324 |
| 4 Computer Science (all) | 313 |
| 5 Computer Networks and Communications | 277 |
| 6 Artificial Intelligence | 272 |
| 7 Theoretical Computer Science | 254 |
| 8 Computer Science Applications | 249 |

| TOP ACADEMIC INST (2015-2019) | #pap | cpp |
|---|---|---|
| 1 Tsinghua University (CHN) | 50 | 2.56 |
| 2 University of Chinese Academy of Scie | 39 | 3.05 |
| 3 Carnegie Mellon University (USA) | 39 | 9.36 |
| 4 Shanghai Jiao Tong University (CHN) | 36 | 7.28 |
| 5 Stanford University (USA) | 31 | 21.29 |
| 6 University of Oxford (GBR) | 29 | 5.41 |
| 7 Chinese Academy of Sciences (CHN) | 29 | 19.14 |
| 8 Duke University (USA) | 26 | 6.69 |
| 9 Peking University (CHN) | 26 | 1.73 |
| 10 University of California at Berkeley (US | 24 | 82.17 |

| TOP SOURCES (2015-2019) | #pap |
|---|---|
| 1 LECT NOTES COMPUT SCI | 244 |
| 2 ADV NEURAL INF PROCES SYST | 113 |
| 3 PROC IEEE COMPUT SOC CONF COMPUT VIS | 87 |
| 4 PROC IEEE INT CONF COMPUT VISION | 39 |
| 5 AAAI CONF ARTIF INTELL, AAAI | 30 |
| 6 IJCAI INT JOINT CONF ARTIF INTELL | 29 |
| 7 PROC INT CONF IMAGE PROCESS ICIP | 25 |
| 8 PROC INT CONF PATTERN RECOGNIT | 24 |
| 9 MM - PROC ACM MULTIMED CONF | 24 |
| 10 IEEE ACCESS | 23 |

| TOP COUNTRIES (2015-2019) | pct | cpp |
|---|---|---|
| 1 USA | 39.53% | 14.55 |
| 2 CHN | 30.45% | 3.57 |
| 3 GBR | 9.75% | 8.37 |
| 4 JPN | 6.66% | 2.15 |
| 5 DEU | 5.81% | 18.85 |
| 6 KOR | 4.18% | 3.94 |
| 7 CAN | 3.94% | 11.29 |
| 8 FRA | 3.03% | 5.94 |

| TOP OTHER INST (2015-2019) | #pap | cpp |
|---|---|---|
| 1 Alphabet Inc. (USA) | 65 | 19.83 |
| 2 Microsoft USA (USA) | 37 | 13.38 |
| 3 Adobe Systems Incorporated (USA) | 34 | 17.09 |
| 4 Facebook Inc (USA) | 28 | 22.64 |
| 5 Tencent (CHN) | 14 | 3.21 |
| 6 Institut National de Recherche en Inforr | 12 | 2.58 |

| TOP SECTORS (2015-2019) | pct |
|---|---|
| 1 Acad | 90.2% |
| 2 Corp | 20.3% |
| 3 Gov | 6.4% |
| 4 Med | 2.3% |
| 5 Oth | 0.8% |

| REPRESENTATIVE PAPERS (2015-2019) | ncited |
|---|---|
| 1 Salimans T. (2016) Improved techniques for training GANs. Advances in Neural Information Processing Systems | 525 |
| 2 Isola P. (2017) Image-to-image translation with conditional adversarial networks. Proceedings - 30th IEEE Conference on C | 512 |
| 3 Denton E. (2015) Deep generative image models using a laplacian pyramid of adversarial networks. Advances in Neural Inf | 428 |
| 4 Zhu J.-Y. (2017) Unpaired Image-to-Image Translation Using Cycle-Consistent Adversarial Networks. Proceedings of the IE | 371 |
| 5 Chen X. (2016) InfoGAN: Interpretable representation learning by information maximizing generative adversarial nets. Adva | 272 |
| 6 Gulrajani I. (2017) Improved training of wasserstein GANs. Advances in Neural Information Processing Systems | 180 |
| 7 Pathak D. (2016) Context Encoders: Feature Learning by Inpainting. Proceedings of the IEEE Computer Society Conferenc | 425 |
| 8 Gatys L.A. (2016) Image Style Transfer Using Convolutional Neural Networks. Proceedings of the IEEE Computer Society ( | 406 |
| 9 Liu Z. (2015) Deep learning face attributes in the wild. Proceedings of the IEEE International Conference on Computer Visic | 532 |
| 10 Johnson J. (2016) Perceptual losses for real-time style transfer and super-resolution. Lecture Notes in Computer Science (i | 467 |

| TOP AUTHORS (2015-2019) | #pap | cpp |
|---|---|---|
| 1 Carin, Lawrence (Duke University) | 19 | 7.74 |
| 2 Henao, Ricardo (Duke University) | 13 | 10.38 |
| 3 Shechtman, Eli (Adobe Systems Incorp | 12 | 26.08 |
| 4 Zhu, Jun (Tsinghua University) | 12 | 6.08 |
| 5 Yang, Ming-Hsuan (University of Califoi | 11 | 12.91 |
| 6 Lee, Honglak (Alphabet Inc.) | 11 | 62.55 |
| 7 Ermon, Stefano (Stanford University) | 11 | 3.45 |
| 8 Hua, Gang (China University of Mining | 10 | 10.10 |
| 9 Yang, Jimei (Adobe Systems Incorpora | 10 | 27.10 |
| 10 Bengio, Yoshua (N/A) | 9 | 20.78 |

| RECENT REVIEWS (2015-2019) | ncited |
|---|---|
| 1 Creswell A. (2018) Generative Adversarial Networks: An Overview. IEEE Signal Processing Magazine | 36 |
| 2 Lin Y.-L. (2018) The New Frontier of AI Research: Generative Adversarial Networks. Zidonghua Xuebao/Acta Automatica S | 1 |
| 3 Wang K. (2017) Generative adversarial networks: Introduction and outlook. IEEE/CAA Journal of Automatica Sinica | 22 |
| 4 Wang K.-F. (2017) Generative Adversarial Networks: The State of the Art and Beyond. Zidonghua Xuebao/Acta Automatica | 38 |
| 5 Kaneko T. (2018) Generative adversarial networks: Foundations and applications. Acoustical Science and Technology | 1 |

| INDICATORS | value |
|---|---|
| # AI PAPERS (2015-2019) | 623 |
| % AI PAPERS (2015-2019) | 37.71% |
| AGE (as of 2018) | 0.54 |
| PAPER VITALITY (as of 2018) | 0.8103 |
| REFERENCE VITALITY (2018) | 0.3056 |
| % INDUSTRY PAPERS (2015-2019) | 20.28% |
| RES LEVEL (2018) | 1.678 |
| Δ RES LEVEL (2013 to 2018) | 0.000 |
| CITES PER PAPER (2015-2019) | 8.38 |
| CITES PER PAPER (2017-2019) | 3.86 |
| PREDICTED GROWTH 2018 | 2.58 |

| TOP CITED DOCUMENTS (2000-2017) | ncited |
|---|---|
| 1 Goodfellow I.J. (2014) Generative adversarial nets. Advances in Neural Information Processing Systems | 3167 |
| 2 Radford A. (2015) Unsupervised representation learning with deep convolutional generative adversarial networks. Unsuper | 1195 |
| 3 Kingma D.P. (2013) Auto-encoding variational bayes. Auto-encoding Variational Bayes | 914 |
| 4 Isola P. (2016) Image-to-image translation with conditional adversarial networks. Image-to-image Translation with Conditior | 554 |
| 5 Liu Z. (2015) Deep learning face attributes in the wild. Proceedings of the IEEE International Conference on Computer Visic | 532 |

**Fig 4. Characterization of STS5 topic #5495.**

there is an underlying assumption that the research environment is predictable. Forecasts assume predictability. In contrast, foresight and scenarios studies tend to be used when there are many possibilities with extremely low probabilities. But instead of arguing whether specific areas of research are predictable or not, we suggest that a high CSI score for a discipline is strong evidence of historical predictability. Low CSI scores may help to identify areas that have low predictability and might best be addressed using foresight or scenario analysis. Overall, the predictability of growth of any specific RC is an assumption that must be looked at from a

critical perspective. Predictability in the past does not guarantee predictability in the future. Nevertheless, large-scale studies of where predictability seems high (or low) can provide fundamental insights into this question.

Predictability also is at the basis for the choice of a three-year forecast window. Bibliometric models are not designed for very short-term forecasts (less than two years out) because of the inherent time delay of publications. Nor do we think they will be as effective for long term forecasts (more than five years) because the underlying structure of research appears to change more rapidly than one would expect (a phenomenon we are looking at separately). The current sweet spot is in the three- to four-year forecast window. Expanding this model to four years is justified when we can reach the goal of a 25% CSI score. Expanding the applicability of these models beyond four years requires far more understanding about how research communities actually evolve.

The methodological weaknesses of this study can be summarized around issues of data, algorithms, indicators, and application. Forecasts can only be made with the data available, and any biases in the data (e.g., by language, nationality, completeness) naturally bias the resulting forecasts. From a database perspective, we have used one of the largest curated bibliographic databases. This helped to simplify a great deal of the pre-processing work that is sometimes needed to create a truly global model of research from the scientific and technical literature. But every database has gaps which need to be kept in mind when creating and evaluating forecasts. For example, an analysis of the field of Artificial Intelligence might be best served by including Chinese language technical literature. In Scopus, China is publishing roughly the same number of articles in this field as is the United States, yet the papers from China are cited much less than those from the United States. Might this citation gap be due to the possibility that the English-based technical literature is highly represented in Scopus while Chinese-based technical literature is not?

Clustering algorithms have advanced a great deal over the past two decades, and while they tend to enable larger and more comprehensive calculations as they evolve, their effect on the accuracy of RCs is hard to quantify. One methodological weakness, therefore, is validating that these RCs are, in fact, being identified more accurately as algorithms advance. We have published extensively on accuracy [cf. 18, 30, 31], yet those studies have almost exclusively investigated relatedness measures rather than algorithmic effects.

Perhaps the biggest flaw in this study is in the indicators that are available for use. The most obvious indicators worked well—those based on publication trends are the most effective at predicting exceptional growth three years in advance. Yet indicators that draw more from an understanding of the life cycle of a research community are noticeably missing. Prior literature has focused on emergence and growth. But when an RC does not keep growing it means that more researchers are leaving the community than entering it. Why do researchers stop entering a community at their previous rates? Why do they leave? These are key phenomena that we know little about and have no indicators by which to predict when a research community is transitioning into maturity or even decline. Specifically, we haven't looked at the age of new entrants or the age of the researchers that stopped publishing in a research community. Signals that might anticipate mass entry or mass exit of researchers into a research community are noticeably missing and are a promising area for future research.

There is also the possibility of bias from an ex post facto perspective. We do not believe the ex post facto perspective has influenced our choice of clustering algorithms, which were selected to improve accuracy [18]. An ex post facto perspective has influenced our choice of indicators (i.e. they "worked" in the DC5 model). But choices in indicators, or in the combinations of indicators, were not based on identifying actual cases of exceptional growth and then

"working backwards" to determine what would have identified them. Nor have we tried to optimize the CSI scores.

Finally, we need a better understanding of what makes a forecast actionable. Our working hypothesis is that forecasts based on positive relative years (forecasts made at or after the model year) are the most actionable because they don't include future information, and that forecasts based on negative RY are circumstantial but may not be actionable due to the leakage of future information. We've included the CSI scores of circumstantial forecasts to show CSI trends that are useful for understanding how the models work over time. One can expect, with better clustering algorithms and document assignment algorithms, that the newer STS5 model will outperform the older DC5 model. But we simply don't know this to be true with the evidence presented to date.

One potential experiment that could address this issue would be to create a new model using data through year $n$ (e.g., 2013) and then to add annual data sequentially. However, the annual additions would be done in an algorithmically different way than before. The Leiden algorithm has the capability of assigning cluster numbers to existing nodes—to seed the new calculation with results from a previous calculation. Thus, one could create a model through 2013 and then create a separate model through 2014 while assigning the existing papers to their 2013 cluster numbers as a starting point, and so on for subsequent years. Done this way, each year starting with 2013 could be used to provide an actionable forecast because of the way the model would be built sequentially and without the inclusion of future information. Until this experiment is completed, however, or until we add three more years of data to the existing STS5 model, our assessments are incomplete despite the promise inherent in the circumstantial forecasts.

In summary, this study represents a starting point. Despite known flaws, the results to date are promising. Indicators have been identified that do a reasonable job of forecasting future growth, and a composite indicator using four indicators has been developed. The forecast events from the 2014 DC5 model shown in Table 6 are strong evidence that the approach works well in those fields in which it does best. The true analytic value of this approach is at the granular (DC5 or STS5) level where research communities are good representations of scientific topics in the Kuhnian sense. Our hope is that future work will lead to the development of a production-level forecasting system based on models with increased accuracy and robustness. Such a system will generate forecasts that will influence decision-making in a positive way.

## Acknowledgments

We thank Michael Patek for creating the models used in this study. Data that can be used to replicate key results are available at https://figshare.com/articles/Data_for_A_Novel_Approach_to_Predicting_Exceptional_Growth_in_Research_/12241727 (Data DOI: 10.6084/m9.figshare.12241727).

## Author Contributions

**Conceptualization:** Richard Klavans, Kevin W. Boyack, Dewey A. Murdick.

**Data curation:** Richard Klavans, Kevin W. Boyack.

**Formal analysis:** Richard Klavans, Kevin W. Boyack.

**Funding acquisition:** Dewey A. Murdick.

**Investigation:** Richard Klavans, Kevin W. Boyack.

**Methodology:** Richard Klavans, Kevin W. Boyack, Dewey A. Murdick.

**Project administration:** Richard Klavans, Dewey A. Murdick.

**Resources:** Kevin W. Boyack.

**Software:** Richard Klavans, Kevin W. Boyack.

**Supervision:** Richard Klavans, Dewey A. Murdick.

**Validation:** Richard Klavans, Kevin W. Boyack.

**Visualization:** Kevin W. Boyack.

**Writing – original draft:** Richard Klavans, Kevin W. Boyack, Dewey A. Murdick.

**Writing – review & editing:** Richard Klavans, Kevin W. Boyack, Dewey A. Murdick.

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
