## [Decision Letter · Decision Letter 0]

16 Jul 2020

PONE-D-20-13116

A Novel Approach to Predicting Exceptional Growth in Research

PLOS ONE

Dear Dr. Boyack,

Thank you for submitting your manuscript to PLOS ONE. After careful consideration, we feel that it has merit but does not fully meet PLOS ONE’s publication criteria as it currently stands. Therefore, we invite you to submit a revised version of the manuscript that addresses the points raised during the review process. Please submit your revised manuscript by Aug 30 2020 11:59PM. If you will need more time than this to complete your revisions, please reply to this message or contact the journal office at plosone@plos.org. Please include the following items when submitting your revised manuscript:

We look forward to receiving your revised manuscript.

Kind regards,

Lutz Bornmann

Academic Editor

PLOS ONE

Journal Requirements:

2. Thank you for inlcuding your competing interests statement; "The authors have declared that no competing interests exist."

We note that one or more of the authors are employed by a commercial company:

SciTech Strategies, Inc

Reviewers' comments:

Reviewer's Responses to Questions

**Comments to the Author**

1. Is the manuscript technically sound, and do the data support the conclusions?

Reviewer #1: Yes

Reviewer #2: Partly

2. Has the statistical analysis been performed appropriately and rigorously? 

Reviewer #1: Yes

Reviewer #2: Yes

3. Have the authors made all data underlying the findings in their manuscript fully available?

Reviewer #1: Yes

Reviewer #2: Yes

4. Is the manuscript presented in an intelligible fashion and written in standard English?

Reviewer #1: Yes

Reviewer #2: Yes

5. Review Comments to the Author

Reviewer #1: The paper presents a study on the development of a method for predicting exceptional growth in research topics, that is, emerging topics. This issue is of high importance to science and technology policy makers, yet there is no widely accepted automatic method available.

A clearly raised limitation is that the construction of research communities relies on "future" information in that the clustering of papers uses information only available after publication, namely the citations from later papers. However, this issue is carefully considered in the analysis and the reasons for including the parts of the analysis based on "future" data are sound. Several other concerns are pointed out, such as algorithmic clusters without a specific topic, which are left for future studies.

Based on the presented method, the paper also gives a specific prediction on likely candidate emerging topics in AI. This is especially commendable, as it allows independent external validation on whether these topics indeed will show the expected exceptional growth in the years to come

Comments:

The authors keep using the terms "classification" and "classify" (i.e. page 7, lines 146, 148 and 150), which are not technically the correct terms when clustering is meant. Classification always involves a priori defined classes.

p. 13 and 14: The definition of the concepts Forecast Year, Target Year and Peak Year should be given before Publication Share and Growth, as Growth Rate relies on them.

Reviewer #2: The following technical concerns should be considered in the revision:

- Expost facto perspective: A big problem of such prediction models in bibliometrics is that they are almost always based on given data from the past. It cannot be excluded that the results are based on specific implicit selections in the light of later results. Such selection biases can manifest itself in the choice of the RC cluster solution or the choice of the predictor variables or the choice of parameters in the optimization process. In my opinion, the problem of such potential biases should be formulated at least as a further limitation of the results of the study. One possible argument against it might be the big sample size.

- Term “research community” (RC): The term "research community" was adopted by Kuhn. I wonder if a grouping of publications according to keywords, citations and references really reflects what Kuhn understands by a "research community" as a cluster of concept, practices, norms and standards. It would certainly be helpful to include Kuhn's definition of RC in the manuscript or to omit it altogether.

- Identification of RC as a black box: The VOS algorithm for identification of RCs and scientific fields is certainly an interesting alternative to journal-based field identification. Nevertheless, it remains unclear, even in this manuscript, what these RCs mean and by which criteria the cluster solution was evaluated. Why 100,000 and not 10,000 RCs. The VOS algorithm is more or less a black box. It would be helpful if the revision could give a little more information about how the RCs came about numerically. As far as I am understand the manuscript correctly the clustering with the VOS algorithm was done for all publications in the respective time frame (e.g., 1996-2012) and not for each year. I wonder whether the clustering might correlate with the frequency pattern of publication. It must be guaranteed that the RCs are independent from the special growth trends over time.

- Critical Success Index (CSI) (p.6): The concepts of false positive, false negative... have strong roots in medical diagnostics. Here the concepts of sensitivity (as ability of a test to correctly classify an individual as ill) and specificity (as ability of a test to correctly classify an individual as not healthy) are of central importance. The concept of CSI is similar to, but not identical with the concept of a positive predictive value (PPV), the percentage of patients with a positive test result who are actually ill. I wonder why the manuscript does not use common concepts that have a clear meaning instead of CSI. It is not fully clear, what CSI actually means. It is also not clear what is meant by a "false positive rate of 67%". Have 67% of RCs without exceptional growth been predicted as exceptional growth?

- Arbitrary threshold values (p. 15): The manuscript contains a number of thresholds without further explanations. One gets the impression of arbitrarily chosen values. Exceptional growth rate is binary defined with 1 if GR_FY>1.08 and 0 if it does not. Why 8% growth not 5% or 10%? Perhaps, a percentile approach might be more adequate (e.g., in 20% percentile of all RCs). RCs were considered with at least 20 papers, why 20 papers? On page 23 a z-test value of 4.0 (probably based on the standard error of the regression coefficient) was mentioned as a criterion for including a variable in the model (stepwise regression), which is rather high and corresponds to an alpha value that exceeds .0001. This might make sense in the case of these large samples, but there is no explanation for this value in the manuscript. A 3-year forecast was justified because “it may present opportunities for action”. In my view short explanations are necessary to justify these thresholds.

- Prediction accuracy: On page 37 it was stated “These four variables very extremely effective in predicting exceptional growth with a pseudo-R2 (28) of 37%”. I constructed a small data example of a 2x2 table (x=binary predictor , y = binary exceptional growth) (50, 10, 10, 50) and calculated a probit regression. The pseudo R_2 was .38 (comparable to the Pseudo-R2 in Table 5). Although this is a strong effect, the corresponding values for sensitivity and specificity are around 83%. These are high values, that’s true, but with a 17% error proportion not an “extremely effective prediction”. The sensitivity of the test for Corona is about .98, the specificity is about .95.

With respect to the criteria of publication of PlosOne the following statements can be formulated. The study represents the results of original work. Results reported have not been published elsewhere. Experiments, statistics, and other analyses are performed to a high technical standard, but are not fully described in sufficient detail. Conclusions are presented in an appropriate fashion, but it somewhat questionable, whether the conclusions (predictors) are fully supported by the data. The article is presented in an intelligible fashion and is written in standard English. The research meets all applicable standards for the ethics of experimentation and research integrity.

6. PLOS authors have the option to publish the peer review history of their article (what does this mean?). If published, this will include your full peer review and any attached files.

Reviewer #1: No

Reviewer #2: No

---

## [Author Response · Author response to Decision Letter 0]

28 Jul 2020

Response to Reviewers

We appreciate the thoughtful efforts of the reviewers and their comments. These have helped us to clarify things that would have raised questions to many readers. We have added and revised multiple elements to the manuscript to provide the additional reasoning and clarity requested. Detailed responses to each point by the reviewers are given below.

Reviewer 1:

Misuse of the word ‘classification’ and ‘classify’

Response: These words were replaced by the word ‘clustering’ (instead of classifying) on page 7,8. As you point out, the clustering approach is creating the classification system. Once the classification system is created, papers (published after the year that the model is created) are assigned to the classification system. 

Definitions (Forecast Year, Target Year and Peak Year) should be moved

Response: These have been moved as requested.

Reviewer 2:

expost facto perspective

Response: While we agree that this can be a problem, we took great pains to avoid the specific examples you mention. 

• Choice of the RC cluster solution: the method we use for clustering documents is based on improved accuracy- as measured using review papers or grant-literature links as the gold standard. We published how we do this- it’s an active (and somewhat contentious) area of research by bibliometricians. We have not looked at choosing different RC cluster solutions in terms of the ‘exceptional growth’ communities that they might create (or that we would expect them to create). The identification of these high growth communities is a relatively recent area of inquiry. 

• Choice of the predictor variables: This is an area where we were most concerned about expost facto perspective. We created our predictor variables solely on the 2013 (DC5) model- knowing what the outcomes actually were. We don’t know the outcome of the STS model- the outcome of our predictions are yet to be determined. The purpose of writing this up during this interim period is to open to the possibility that the predictor variables won’t work. We won’t have an answer to that question for two years. And even when we get an answer to that question- it will be subject to the way in which articles, published in 2019, 2020 and 2021, are assigned. 

• Choices of parameters in the optimization process: We did not actually use an optimization process. An optimization process would have looked at different coefficients for the predictor variables that would maximize the CSI score. We choose not to do this for the reason you mention. At this point in our investigation- our goal isn’t to maximize the CSI score. It’s to show that a reasonable combination (derived from the probit analysis) would result in a sufficiently good result to warrant further investigation.

• Some of the above explanation has been incorporated in the document (lines 826+), some has not.

The term ‘Research Community’

Response: We’ve modified the text (lines 169+) to include a quote and the page numbers that will lead the reader to a more thorough definition of ‘Research Community’ (and how it differs from a ‘paradigm’). There are two pages (page 178 and 179) in the 2nd Edition of his book (Structure of Scientific Revolutions) that are particularly instructive. His suggestions- on how to identify a research community- are explicit. His suggestions do not include looking at keywords or journals. He explicitly mentions Garfield’s technique of direct citation analysis in this discussion. 

RC is a black box. How is a research community evaluated? Why 100,000 (vs. 10,000 or 1000) clusters? Is the clustering algorithm independent of growth trends? (see comments re: Kuhn (page 8). 

Response:

• Reference #18 (Klavans & Boyack, 2017) describes one way to evaluate research communities- and provides strong evidence that journal-based methods are far less accurate than the method we are using. That article also creates solutions at the 100,000, 10,000, 1000 and 100 level- and explains why we choose 100,000 to characterize research communities. 

• One can also arrive at the 100,000 level by building on Kuhn’s observation that the average research community has about 100 people (mentioned on page 178-179). With roughly 10,000,000 researchers world-wide- you come up with about 100,000 research communities. Kuhn mentions (again on pages 178 and 179) the difference between fields, disciplines, specialties and research communities. We operationalize this in terms of setting parameters (for the number of clusters) at 102; 103; 104 and 105. 

The VOS algorithm is more or less a black box. It would be helpful if the revision could give a little more information about how the RCs came about numerically. As far as I understand the manuscript correctly the clustering with the VOS algorithm was done for all publications in the respective time frame (e.g., 1996-2012) and not for each year. I wonder whether the clustering might correlate with the frequency pattern of publication. It must be guaranteed that the RCs are independent from the special growth trends over time.

Response: The VOS and SLM (and Leiden) algorithms (see references 15-17) are modularity-based algorithms, and modularity-based clustering algorithms (including the Mark Newman’s original code and the Leuven code) are well accepted across many disciplines. These algorithms are not black boxes but operate on the principle of maximizing the ratio of links within cluster to those outside clusters. Results are, of course, dependent on the network being submitted to clustering. RCs reflect partitions within the network. To the extent that frequency patterns or growth trends are reflected in the network, they will rightfully be part of RCs.

Critical Success Index: why not use the terms more commonly used in medical diagnostics? It is not clear what CSI is measuring. For example, what is meant by a false positive?

Response: We have elaborated (lines 125-146) on why we chose CSI and what is being measured. We did not (and will not) use terms that are common in medical diagnostics because the analysts and decision makers are not medical diagnosticians. We settled on CSI because it is equivalent to a concept used by analysts and decision-makers concerned about the impact of severe weather. As now explained in the text- predicting exceptional growth in three years is similar in difficulty to predicting the likelihood of a major storm in three days. No-one expects 98% accuracy (as you mention in your next comment about a Covid diagnostic). The ‘state of the art’ for a 3-day forecast in weather forecasting is about 25%. A graph showing predictive accuracy in weather forecasting for 1, 2 and 3 day forecasts can be found at 

https://www.e-education.psu.edu/files/meteo410/image/Lesson3/threat_score0202.html

In this context, a false positive is a prediction of exceptional growth that didn’t occur (analogous to a prediction of bad weather that didn’t occur). 

Arbitrary Threshold Values requires short explanations 

Response: We have provided short explanations for each of the thresholds you identified. 

• 8% relative growth threshold line 320+)). 

o While the 8% value is arbitrary, it is based on a 50 year tradition in business portfolio analysis to use a 10% annual growth threshold (in actual revenue) to distinguish between different market opportunities (‘stars’ and ‘question-marks’ are above 10%; ‘cash cows’ and ‘dogs’ are below 10%). The history (and popularity) of this approach can be found at the following site: https://www.bcg.com/en-us/about/our-history/growth-share-matrix. When this was first introduced, they choose a 10% threshold- simply because it ‘looked good’. Since we are using growth in publication share (not actual growth in publication numbers), and the average publication growth rate is roughly 2% annually, using an 8% annual growth rate in publication share is the same as a 10% annual growth rate in actual publication levels. 

o The 8% value is also based on feedback from users. Most users want to evaluate research communities that are opportunities- including those that might be more ‘on the margin’. Again- this is rooted in practice- there is no ‘optimal’ number to use nor have we done any sensitivity analysis to determine which growth rate might yield better prediction scores. As stated previously, our goal isn’t to maximize the CSI score- it’s to find a viable approach and report our initial findings.

o The 8% growth rate in publication share was also based on an examination of the distribution of this variable. Basically- any threshold between 2 and 3 standard deviations from the mean could have been used (beyond 3 SD the results become less relevant to users- they want to see more possibilities). We decided not to include this reasoning in the article because it raises more issues that it resolves. But underlying this choice was an examination of the underlying distribution of the growth statistic.

• RCs with 20 or more papers. The discussion of this issue (line 339+) initially mentioned the following reason for this threshold. We had noticed that there were a very large number of very small communities that ‘disappear’ in a few years (a new model doesn’t keep the papers together). We initially thought this was due to information leakage. We have revised our paper based on the following.

o We have been investigating the phenomena of ‘disappearing small communities’ and have discovered that it’s because the majority of the very small research communities have extremely few inter-citations. These are ‘loosely defined’ communities (which might, in Kuhn’s world, define them as ‘pre-paradigmatic’). We will be able to create a more rigorous threshold in the future base on follow-up work on how network characteristics (such as density) affect community evolution. The working hypothesis is that pre-paradigmatic communities cannot have exceptional growth. 

o The more important reason (also added to the manuscript) is this study is meant to inform practice rather than to be academic. Users (policy makers, corporations, funders, etc.) don’t really care about exceptional growth if the community is very small. They care about potential impact which can be roughly modeled as mass (the number of papers) times velocity (which correspond to the indicators we are using to predict exceptional growth). A community size of 20 is about the lowest amount that we’ve observed any user as being concerned about. 

• Z test value of 4 or more (line 508+)

o The reasons for this threshold are pragmatic. We are trying to create a model that is as simple as possible (the fewest number of variables) and stable as possible (the variables are unlikely to be replaced when new variables are created and tested). To accomplish this, we reject variables that might be significant- but don’t have a sizable increase in adjusted R-square. We’ve used a Z test value of 4 as a way of measuring resilience. It’s been our experience that variables with a Z test less than 4 are not stable candidates when new variables are developed. 

• three-year forecast (line 782+)

o The three year threshold is based on assumptions about the effective planning horizon of the modeling technique. Bibliometric models are not designed for very short term forecasts (less than 2 years out) because of the inherent time delay of publications. Nor do we think they will be as effective for long term forecasts (more than 5 years) because the underlying structure of research appears to change more rapidly than one would expect (a phenomenon we are looking at separately). The current sweet spot is in the 3 to 4 year period. Expanding the applicability of these models beyond 4 years requires far more understanding about how research communities actually evolve. 

Prediction Accuracy 

Response: Text was added on page 6 (mentioned above). We agree that the prediction accuracy, while high, does not meet the requirements of a diagnostic test. Sensitivity and specificity are nowhere near the 98% and 95% level required for detecting Covid. But we re-iterate, this is not the threshold that is needed for effective threat assessment in preparing for a major storm. There is certainty about whether someone has Covid- so we can expect a diagnostic test to reach high levels of certainty. But no-one expects weather forecasters to be that accurate- it’s simply not in the nature of weather to be that predictable. Nor is this the threshold needed for effectively responding to a major opportunity in research. It’s simply not in the nature of research to be that predictable. Even forecasts of mortality rates from being infected by the Covid virus aren’t that accurate- we knew so little about this unique virus when it first appeared. We re-iterate that the methods used in medical diagnostics deal with conditions where extremely high levels of accuracy are possible. They are not as applicable to the types of situations we are focusing on.

---

## [Decision Letter · Decision Letter 1]

2 Sep 2020

A Novel Approach to Predicting Exceptional Growth in Research

PONE-D-20-13116R1

Dear Dr. Boyack,

We’re pleased to inform you that your manuscript has been judged scientifically suitable for publication and will be formally accepted for publication once it meets all outstanding technical requirements.

Kind regards,

Lutz Bornmann

Academic Editor

PLOS ONE

Additional Editor Comments (optional):

Reviewers' comments:

Reviewer's Responses to Questions

**Comments to the Author**

1. If the authors have adequately addressed your comments raised in a previous round of review and you feel that this manuscript is now acceptable for publication, you may indicate that here to bypass the “Comments to the Author” section, enter your conflict of interest statement in the “Confidential to Editor” section, and submit your "Accept" recommendation.

Reviewer #1: All comments have been addressed

Reviewer #2: All comments have been addressed

2. Is the manuscript technically sound, and do the data support the conclusions?

Reviewer #1: Yes

Reviewer #2: Yes

3. Has the statistical analysis been performed appropriately and rigorously? 

Reviewer #1: Yes

Reviewer #2: Yes

4. Have the authors made all data underlying the findings in their manuscript fully available?

Reviewer #1: Yes

Reviewer #2: Yes

5. Is the manuscript presented in an intelligible fashion and written in standard English?

Reviewer #1: Yes

Reviewer #2: Yes

6. Review Comments to the Author

Reviewer #1: The raised concerns have been adequately addressed. The inaccurate use of the term classification has been remedied. The definitions of the time periods have been moved.

Reviewer #2: I very much appreciate the detailed answers to the reviewers' comments and the changes made to the manuscript. For future analyses and manuscripts I would recommend to use also the concepts of medical test diagnostics (e.g. valid negative, valid positive, basis rate, ...), because they are well known even outside of medicine and help the reader to quickly evaluate a model, even if the figures are significant lower than in medical testing.

7. PLOS authors have the option to publish the peer review history of their article (what does this mean?). If published, this will include your full peer review and any attached files.

Reviewer #1: No

Reviewer #2: No

---

## [Editor Report · Acceptance letter]

7 Sep 2020

PONE-D-20-13116R1 

A Novel Approach to Predicting Exceptional Growth in Research 

Dear Dr. Boyack:

I'm pleased to inform you that your manuscript has been deemed suitable for publication in PLOS ONE. Congratulations! Your manuscript is now with our production department. 

Kind regards, 

on behalf of

Dr. Lutz Bornmann 

Academic Editor

PLOS ONE